# The Effect of Mushroom Culture Filtrates on the Inhibition of Mycotoxins Produced by *Aspergillus flavus* and *Aspergillus carbonarius*

**DOI:** 10.3390/toxins15030177

**Published:** 2023-02-25

**Authors:** Jelena Loncar, Barbara Bellich, Paola Cescutti, Alice Motola, Marzia Beccaccioli, Slaven Zjalic, Massimo Reverberi

**Affiliations:** 1Department of Ecology, Aquaculture, and Agriculture, University of Zadar, 23000 Zadar, Croatia; 2Department of Environmental Biology, Sapienza University of Rome, 00185 Rome, Italy; 3Department of Life Sciences, University of Trieste, 34127 Trieste, Italy

**Keywords:** aflatoxin B_1_, ochratoxin A, *Schizophyllum commune*, schizophyllan, biological control

## Abstract

Two of the mycotoxins of greatest agroeconomic significance are aflatoxin B_1_ (AFB_1_), and ochratoxin A (OTA). It has been reported that extracts from some wood-decaying mushrooms, such as *Lentinula edodes* and *Trametes versicolor* showed the ability to inhibit AFB_1_ or OTA biosynthesis. Therefore, in our study, a wide screening of 42 isolates of different ligninolytic mushrooms was assayed for their ability to inhibit the synthesis of OTA in *Aspergillus carbonarius* and AFB_1_ in *Aspergillus flavus*, in order to find a metabolite that can simultaneously inhibit both mycotoxins. The results showed that four isolates produce metabolites able to inhibit the synthesis of OTA, and 11 isolates produced metabolites that inhibited AFB_1_ by >50%. Two strains, the *Trametes versicolor* strain TV117 and the *Schizophyllum commune* strain S.C. Ailanto, produced metabolites able to significantly inhibit (>90%) the synthesis of both mycotoxins. Preliminary results suggest that the mechanism of efficacy of the *S. commune* rough and semipurified polysaccharides could be analogous to that found previously for Tramesan^®^, by enhancing the antioxidant response in the target fungal cells. The overall results indicate that *S. commune’s* polysaccharide(s) could be a potential agent(s) in biological control and/or a useful component of the integrated strategies able to control mycotoxin synthesis.

## 1. Introduction

Mycotoxins are low-molecular-weight secondary metabolites synthesized by some filamentous fungi that are toxic to vertebrates [1,2]. The effects of mycotoxins on humans and animals vary according to their chemical structure and concentration and can cause acute and/or chronic effects. Cytotoxicity is the main cause of acute mycotoxicosis, while genotoxicity, which in some cases can have carcinogenic effects, is connected to chronic mycotoxicosis [2,3,4]. Often, more than one mycotoxin is found on contaminated food and feed commodities. The research on possible synergic effects of co-contamination with different mycotoxins is still ongoing [2,5,6], although, the synergistic effects of mycotoxins have not been sufficiently studied to date. Aflatoxins (AF) B_1_, B_2_, G_1_, and G_2_, as well as ochratoxin A (OTA), are among the most important and studied mycotoxins. Aflatoxin B_1_ (AFB_1_) is classified as carcinogenic (group 1A) by the International Agency for Research on Cancer [7] while OTA is classified as a possible carcinogen for humans, group 2B [8]. Aflatoxin B_1_ and ochratoxin A are important in a range of economically important commodities. Both mycotoxins are synthesized by species of the genus *Aspergillus* which are xerophilic and able to grow under water stress conditions, much wider than those allowing plant growth. They can thus colonize and contaminate a range of food commodities both in the pre-and postharvest period [9]. Moreover, both AFB_1_ and OTA are stable molecules and relatively heat resistant during food and feed processing [10]. Furthermore, both can be transmitted along the food chain and stored in some tissues of animals fed with contaminated feed [11]. Due to the severity of the toxin, the presence of approximately a dozen of mycotoxins, including AFB_1_ and OTA, are regulated by EC regulation [12].

Different strategies to control the presence of mycotoxins in food and feed have been applied for many years, however, none of them have been completely successful. Thus, there is still a need for minimization and prevention control strategies for these mycotoxins. Many strategies have been predominantly based on the use of chemical compounds (pesticides and fungicides). The number of chemicals that were applied in the prevention of fungal growth and/or mycotoxin control has been banned by authorities [13,14,15]. This, together with rising environmental awareness of the population, led to an increase in the research for new more environmentally friendly strategies in mycotoxin control. Detoxification methods can be applied to contaminated seeds, and several physical, chemical, and biological methods are under research. Moreover, some interesting results were obtained by using curcumin and purified enzymes, including laccases, manganese peroxidase, and the recently identified bacillus aflatoxin-degrading enzyme [16,17,18,19,20,21].

In the last 20 years, different research confirmed that mushroom polysaccharides could represent a valid tool in mycotoxin control [22,23,24,25,26]. The most studied inhibiting effects of mushroom metabolites are those made with *Lentinula edodes* [22] and *Trametes versicolor* [15,24,25] using lyophilized culture filtrates. Moreover, Scarpari and collaborators [25] were able to identify the active principles produced by *T. versicolor*, which was an exopolysaccharide released in its culture filtrate, called tramesan. tramesan can act as a pro-antioxidant in different organisms and has biological activity in blocking AFB_1_ and OTA production probably by acting on gene expression and consequent stimulation of antioxidant activity in the fungal cell [22,23,24]. Furthermore, Loncar and collaborators [15] have demonstrated that tramesan fragments larger than seven units inhibited up to 90% AFB_1_ synthesis by *A. flavus* and up to 81% of the biosynthesis of OTA by *A. carbonarius.* Even if tramesan is a very promising tool in the challenge against the presence of mycotoxins in food and feed, the production costs could be too high for large-scale use. Moreover, tramesan is produced by a living organism and some mutation(s) could alter its structure and efficacy. Given the similar mechanism of action of the mushroom polysaccharides studied so far, the present study screened 42 isolates of different ligninolytic mushrooms present in the collection of the Laboratory of Plant Pathology and Mycology of the Sapienza University of Rome for their capacity to produce metabolites able to control simultaneously the biosynthesis of both AFB_1_ and OTA. The goal was to identify the tool that could be effectively used for the significant control of AFB_1_ and OTA production and use in food and feed applications. Furthermore, the aim was to clarify the structure of the mushroom polysaccharides capable of inhibiting the synthesis of mycotoxins, so that it could be established whether there is a common structure that can be used in designing synthetic glucans for mycotoxin control.

## 2. Results

### 2.1. Wide Screening Mushrooms Assay

This research aimed to identify mushroom polysaccharides able to inhibit the synthesis of both AFB_1_ and OTA. The first step was a wide screening assay of lyophilized liquid culture filtrates (LFC) of the different isolates of different mushroom species. The inhibition of OTA and AFB_1_ biosynthesis in the presence of 2% *w/v* of LCF of different isolates of ligninolytic mushrooms expressed as a percentage compared to nontreated samples (control) are shown in Table 1. For the purpose of this study, only inhibition rates higher than 50% were considered significant.

The results indicate that the inhibition of both mycotoxins was more isolate-dependent than species-dependent. This is particularly visible in the case of different isolates of *G. adspersum*. One isolate of *G adspersum*, COLLETTA 12, showed significant inhibition (*p* < 0.001) of both mycotoxins, however, the inhibition was lower than 50%, and therefore considered of no significance for the scope of this research. P COLLEGNO 2A inhibited more OTA biosynthesis than AFB_1_. One, Tiglio, inhibited the biosynthesis of both mycotoxins for about 30%, and the LCF of four isolates showed inhibition of AFB_1_ synthesis over 50% while the inhibition of OTA was under 50% (Figure 1). A similar situation can be observed for *S. commune* isolates. Two isolates, DP 61 and GTM8 R1 showed low inhibition rates for both toxins. Isolate BELLAGIO 2 showed inhibition rates between 40 and 50% for both toxins while lyophilized filtrate of isolate S. C. Alianto inhibited the biosynthesis of both mycotoxins for over 90% (Figure 2). The yield of LCFs doesn’t seem to be directly correlated with the inhibition of AFB_1_ and OTA. Different isolates of some species, like *Fomes fomentarius*, show minor variations in LCF yield, while for the others (i.e., *Ganoderma adspermum* and *G. resinaceum*), the variation in the yield was higher (Table 1). The aim of this research was to isolate a mushroom polysaccharide able to provide a high inhibition of both mycotoxins and compare it with tramesan. Therefore the LCF of *Schizophyllum commune* SC Alianto was chosen for further research.

### 2.2. Influence of the LCF of S. commune on AFB_1_ and OTA Present in the Media

To verify that LCF of *S. commune* inhibits mycotoxin synthesis and does not interfere with mycotoxins present in the media (degradation, adsorption, or others), the known concentration of standards of AFB_1_ and OTA in potato dextrose broth (PDB) were incubated for three days in the presence and absence (control) of *S. commune’s* LCF 2% *w/v*. The results (Table 2) show that LCF does not interfere with the toxins present in the media and that the lower concentration of mycotoxins observed in the culture with the presence of LCF is due to the inhibition of the toxin synthesis rather than due to the interference with the mycotoxins present in the culture media.

### 2.3. Assay of the Influence of LCF and Partially Purified pLCF of S. commune S. C. Alianto

LCFs of *S. commune* S. C. Ailanto were partially purified by pure ethanol precipitation and subsequent dialysis (see the Section 5 for details). Obtained partially purified LCF contained mostly polysaccharides and glyco material, and the protein part was further reduced by treatment with proteinase K followed by centrifugation which gave two fractions, namely a supernatant, sLCF, and a precipitate, pLCF. UV spectroscopy of the two fractions showed the absence of absorbance at about 280 nm, thus suggesting that proteins were successfully removed (data not shown). Both fractions were lyophilized and assayed at a concentration of 2% *w/v* on AFB_1_ and OTA production. The sLCF fraction showed no inhibition of mycotoxin synthesis (data not shown) while the effect of pLCF on the synthesis of AFB_1_ and OTA was not significantly different (*p* > 0.001) from that obtained with LCF (Figure 3). The results obtained with pLCF suggested that a polysaccharide (or polysaccharides) might be involved in the inhibition.

### 2.4. Composition Analysis of pLCF and sLCF Fractions

The monosaccharides present in the two fractions were determined by gas–liquid chromatography after methanolysis and derivatization to trimethylsilyl methyl glycosides. The fraction sLCF was found to contain mannose, galactose, and glucose in relative molar ratios of 8:2:1, while the sample pLCF was to be mainly constituted of glucose, with only traces of mannose. 

### 2.5. ^1^H NMR Spectroscopy of pLCF and sLCF Fractions

The ^1^H NMR spectrum of pLCF was recorded at 70 °C in deuterated DMSO: D_2_O in a 6:1 ratio in order to overcome the high viscosity of its water solutions and to eliminate the resonances arising from exchangeable hydroxyl protons. The ^1^H NMR spectrum is shown in Figure 4A. Apart from two intense signals at 2.53 (not shown) and 3.57 ppm attributed to DMSO and residual HOD, respectively, it contains signals typical of polysaccharides. Comparison of the pLCF spectrum with that of the β-glucan Schizophyllan recorded in the same experimental conditions [27] showed that the two samples are extremely similar, if not identical. Resorting to the data in reference [27], the signal at 4.24 ppm, together with the three overlapping resonances between 4.60 and 4.50 ppm, were assigned to the anomeric protons of β-glucose residues, while that at 4.08 was attributed to the H-6 of the branched β-glucose residue. The ring proton resonances included in the range 3.80–3.00 ppm are also in very good agreement with those attributed to Schizophyllan.

The ^1^H NMR spectrum of sLCF, recorded at 50 °C in D_2_O (Figure 4B), showed signals typical of anomeric and ring protons of polysaccharides. In particular, the anomeric region contains many resonances, some of them also overlapped, in agreement with sugars in the α-anomeric configuration. The abundance of signals might be attributed to a mixture of polysaccharides or the presence of a rather complicated and not-so-regular repeating unit, as often found in fungi polysaccharides.

### 2.6. Assay of the Influence of Partially Purified pLCF of S. commune S. C. Alianto and Commercial Schizophylan (SCH) on AFB_1_ and OTA Biosynthesis 

To verify its eventual involvement in mycotoxin inhibition, SCH was assayed for the inhibition of AFB_1_ and OTA. The results show that SCH has some inhibiting effects (about 40%) on both AFB_1_ and OTA, however, this inhibition is significantly (*p* < 0.001) lower than the one observed in pLCF and LCF (Figure 5). 

### 2.7. Mechanism of Action of pLCF

#### 2.7.1. The Activity of Antioxidant Enzymes CAT (a), GPX (b), and SOD (c), during the Time Course in the Mycelium of *A. flavus*

As depicted in Figure 6c, SOD activity was significantly stimulated (*p* < 0.001) by the addition of pLCF in the substrate, in comparison to the control, and a peak was evident at 36 h of incubation. Even if a decrease appeared after this time, the activity of the SOD enzymes remained higher than that in control up to 72 h of incubation. CAT (Figure 6a) activity in the presence of pLCF of *S. commune* was not significantly different (*p* > 0.001) compared to those detected in the control. GPX (Figure 6b) was significantly stimulated (*p* < 0.001) by the presence of pLCF with the highest peak during 72 h of incubation

#### 2.7.2. The Activity of Antioxidant Enzymes CAT (a), GPX (b), and SOD (c), during the Time Course in the Mycelium of *A. carbonarius*

CAT activity was significantly stimulated (*p* < 0.001) by the addition of the pLCF filtrates of *S. commune* in the substrate in comparison to the control (Figure 7a) with the highest peak at 36 h of incubation. After 36 h, the CAT enzyme decreased, and at 72 h of incubation reached the control level. SOD activity was not significantly stimulated (*p* > 0.001) by the addition of the pLCF of *S. commune* in the substrate in comparison to the control (Figure 7c). GPX was significantly stimulated (*p* < 0.001) by the presence of pLCF with the highest peak at 24 h of incubation (Figure 7b). 

## 3. Discussion

Mushroom ꞵ-glucans are widely reported as biologically active compounds and are considered biological response modifiers, or BRM [28,29]. Moreover, fungal β-glucans could be among the factors responsible for the inhibiting effect of aflatoxins [22,23,24], both by acting as free radical scavengers [30] or by stimulating antioxidant enzymes in fungal cells [22,23,24,25]. The wide screening of 42 mushroom isolates showed that different mushroom isolates were able to inhibit, to some extent, the biosynthesis of AFB_1_ and/or OTA, though only two LCFs, *T. versicolor* isolate TV 117 and *S. commune* isolate S.C Ailanto, were able to inhibit the biosynthesis of both mycotoxins, AFB_1_ and OTA, for more than 50%. Therefore it was decided to continue the research only with this strain. Moreover, the results showed the high variability of the inhibiting effects on the mycotoxin synthesis of metabolites produced by different isolates of the same mushroom species, which is consistent with the results reported by Reverberi and collaborators [22,23] and Zjalic and collaborators [24]. In this research, different isolates of *L. edodes* and *T. versicolor* were assayed on aflatoxin inhibition. The observed variability was in any case lower than the one observed in the case of the actual research. The variability in inhibition rates among the different isolates of the same species could depend on the genetic structure of the strain and its capacity to produce inhibiting compound(s). Also, the variability could be a consequence of different environmental adaptations of different strains, and active compounds produced under different environmental conditions such as temperature, pH, and levels of oxygenation [22,24]. Moreover, different types of molecules, some with antioxidant activity per se and others as indirect antioxidants, have been reported as active in AFB_1_ and/or OTA control [17,20,31]. Other compounds have been reported to stimulate the biosynthesis of AFB_1_ [32]. The composition of LCF depends on the isolate’s metabolism and it is a mixture of different compounds, some of which could inhibit mycotoxin synthesis while others could enhance it [22,26,33]. Differences in LCF yield could also be explained by the variations in the metabolism of different strains. 

Results reported in Table 2 confirmed that a putative polysaccharide produced by *S. commune* interferes with toxin synthesis and that it is not involved in the degradation or some kind of masking, of the two toxins. Partial purification of LCF to pLCF highly enhanced the concentration of polysaccharides. The fact that this purification did not reflect on mycotoxin inhibition indicates that polysaccharide(s) is involved in the inhibition. Furthermore, pLCF contains polysaccharides with a higher MM, while sLFC, in which smaller saccharides were present, had no effects on AFB_1_ and OTA inhibition (data not shown), which is in line with the general consideration that high molecular mass glucans appear to be more effective than those of low molecular mass [15,34,35]. 

*S. commune* is known as a producer of bioactive β-glucan, schizophyllan (SCH). SCH is produced during mushroom growth, and it is released in the media in liquid fermentation [36]. Antitumoral and other BRM activities of SCH have been described [29,35]. Its structure is well-known. It is a nonionic homoglucan, with a β-(1-3)-linked backbone and single β-(1-6)-linked glucose side chains at approximately every third residue, generally with an MM of several hundred kDa. It has also been reported that its biological activity depends on its MM as well as its triple helix conformation [35,37]. Composition analysis of pLCF revealed that it is a homoglucan, while ^1^H NMR spectroscopy showed a very high similarity with the spectrum of SCH reported in reference [27]. In order to undoubtedly establish that pLCF has the same primary structure as SCH, further 2D NMR experiments have to be performed. 

It is known that oxidative stress in fungal cells is a trigger for the synthesis of AFB_1_ and OTA [22,31]. Mushroom polysaccharides obtained from *L. edodes* and *T. versicolor* inhibit AFB_1_ and OTA synthesis by activating the fungal antioxidant system and thus preventing oxidative disbalance in fungal cells [15,22,23,24,25,31]. It could be suggested that the S.C. Ailanto pLCF influences the cascade of signals that allows mycotoxin biosynthesis, even if we cannot indicate at which step of the pathway this event occurs. Nevertheless, it could be hypothesized that β-glucans, and other compounds contained in the pLCF filtrate of S.C. Ailanto, could be able to inhibit mycotoxin biosynthesis in *A. flavus* and *A. carbonarius* by enhancing the internal antioxidant system. In particular, pLCF stimulated the activity of SOD and GPX (Figure 6 and Figure 7), which is consistent with the results obtained by Reverberi and collaborators [31] and Zjalic and collaborators [24], in which the activity of these two enzymes was more important for AFB_1_ inhibition than the activity of CAT. The stimulation of enzyme activity involves Apyap1. This protein is activated by the presence of mushroom polysaccharides and enhances the transcription of antioxidant genes in the fungal cell [31,38]. The receptor(s) for mushroom polysaccharides that initiates the cascade is still unknown. Similar mechanisms of action of pLCF and tramesan could suggest that they bind to the same type of receptor and reveling the structure of other S.C Ailanto polysaccharides active in mycotoxin inhibition, which could contribute to a better understanding of this mechanism. 

## 4. Conclusions

In conclusion, the *S. commune* isolate studied in this research could be regarded as a potential agent in biological control or as a useful component among the integrated strategies against mycotoxin-producing fungi in food and feed products. Nevertheless, future research is required to confirm that pLCF is indeed SCH and further determine which specific saccharidic sequence is active in the inhibition of AFB_1_ and OTA so that it can be synthetically produced and used for large-scale applications in agriculture. In particular, the discovery of the very active part of the polysaccharide would be especially relevant, from an economic point of view, as its synthesis could be more cost effective. Moreover, fungal and plant extracts could represent a promising tool in mycotoxin control, compared to chemicals [39,40,41,42,43], due to their lower toxicity when released into the environment. 

## 5. Materials and Methods

### 5.1. Fungal Strains and Mushrooms Used in this Study 

*A. flavus* (Speare) (NRRL 3357), a producer of aflatoxin B_1_, was cultured on Potato dextrose agar (PDA, Himedia, Mumbai, India), at 30 °C for 7 days in dark conditions, and a suspension of 1 × 10^6^ conidia per mL in sterilized distilled water was used as an inoculum in all experiments.

*A. carbonarius*, a producer of ochratoxin A, was isolated by the Laboratory of Plant Pathology and Mycology of the Sapienza University of Rome, Italy. It was grown on PDA at 30 °C for 7 days in dark conditions, and a suspension of 1 × 10^6^ conidia per mL in sterilized distilled water was used as an inoculum in all experiments. All the mushroom isolates (Table 3) were provided by the fungal germplasm bank of the Department of Agricultural, Forest and Food Sciences, University of Turin, Italy, Prof. Paolo Gonthier, and his group.

### 5.2. Production of Mushroom Unrefined Lyophilized Filtrates (LCFs)

The 42 isolates of different mushroom cultures detailed above were kept on a potato dextrose agar (PDA, Himedia, Mumbai, India) medium at 4 °C, and the cultures were subcultured onto fresh medium every 30 days. Three plugs of 1 cm diameter of each isolate were inoculated in sterile conditions in 100 mL of potato dextrose broth (PDB, Himedia, Mumbai, India) and incubated for 15 days at 25 °C under shaken conditions (100 rpm). Then, the liquid cultures were homogenized, in sterile conditions. After homogenization, an aliquot (5% *v/v*) of the fungal cultures was inoculated in 500 mL of PDB in 1L-Erlenmeyer flasks and incubated for 21 days at 25 °C under rotary shaken conditions (100 rpm). After incubation, the mycelia were separated from the culture filtrates by subsequent filtrations through 0.45 μm filters (Sartorius, Goettingen, Germany), to eliminate all the mycelia. Mycelia-purified culture filtrates were concentrated in rota-vapor (Rotavapor^®^ R-300, Buchi, Essen City, Germany), and up to a concentration of 100 mL as described by Parroni and collaborators [42]. Concentrated culture filtrates were lyophilized and utilized for subsequent analyses as reported in [24]. Finally, concentrations of 2% *w/v* of each mushroom exopolysaccharide unrefined substrate were utilized for the experiment. 

### 5.3. Yield

To calculate the yield of LCF, one L of culture filtrates was reduced in volume 10 times in rotavapor at 45 °C, and 50 mL of the reduced LCF were placed in glass backers previously weighed (after being dried for 48 h at 80 °C in thermostat) and lyophilized. After the lyophilization, the backers were weighed again and the DW of LCF was calculated.

### 5.4. In Vitro Screening of Mushroom Polysaccharides on Inhibition of Mycotoxin Synthesis by Aspergillus flavus and Aspergillus carbonarius

For the purpose of wide screening of mushroom polysaccharides, the microtiter-based bioassay using 96-well microplates was performed. This assay allowed the testing of all the fractions using small quantities of the compounds with effective replication (*n*-10) over short periods of time, as previously described by Loncar and collaborators [15].

Screening assays were carried out with 100 conidia of each strain (*A. flavus* or *A. carbonarius*), independently, in 10 μL of sterilized distilled water. This was inoculated together with 190 μL of PDB, in the presence or absence of lyophilized mushroom filtrates in each of the wells. A nutrient medium (PDB) inoculated with the fungus *A. flavus* or *A. carbonarius* was used as a control. The microtiter plates were incubated at 30 °C for 3 days in dark conditions. Different cultures were independently filtered with Millipore filters (0.22 μm, Sartorius, Goettingen, Germany). The filtrate was subsequently used for mycotoxin extraction. 

### 5.5. Mycotoxins Extraction and Quantification 

#### 5.5.1. Ochratoxin A 

The different cultures were independently filtered through Millipore filters (0.22 μm, Whatman, Merck, Darmstadt, Germany). The extraction of OTA was performed, for each condition (control and treated with 2% *w/v* of different mushroom polysaccharides), using the extraction solution of Acetonitrile: Water: Acetic acid (79:20:1 *v/v*), with the addition of 5 µL of Quercetin (≥95% Sigma-Aldrich, Merck, Darmstadt, Germany) (100 µM) as an internal standard, as previously reported in Fanelli and collaborators [44]. The mixture was vortexed for 1 min, centrifuged, and then the lower phase was drawn off. The extraction was repeated three times and the samples were concentrated under a N_2_ stream and redissolved in 50 μL of methanol. The concentration of OTA was determined by HPLC/MS (Agilent, Waldbronn, Germany) and expressed in ppb as described by Amezqueta and collaborators [45].

#### 5.5.2. Aflatoxin B_1_

The different cultures were independently filtered through Millipore filters (0.22 μm, Whatman, Merck, Darmstadt, Germany). The extraction of AFB_1_ was performed for each condition (control and treated with 2% *w/v* of different mushroom polysaccharides), using the extraction solution of Chloroform (VWR, Randor, PA, USA): Methanol (VWR, Randor, PA, USA) (2:1 *v/v*) with the addition of 5 µL of Quercetin (≥95% Sigma-Aldrich, Merck, Darmstadt, Germany) (100 µM) as an internal standard. The mixture was vortexed for 1 min, centrifuged, and then the lower phases were drawn off. The extraction was repeated three times and the samples were concentrated under a N_2_ stream, then redissolved in 50 μL of methanol. The concentration of AFB_1_ was quantified by HPLC/MS (Agilent, Waldbronn, Germany) and expressed in ppb, as described by Fanelli and collaborators [44].

### 5.6. Influence of LCF on Mycotoxins Already Present in the Culture

To verify whether LCF has any influence on mycotoxins already present in the culture media, a solution of PDB containing 10 μg L^−1^ of AFB_1_ or OTA was prepared and 2% (*w/v*) of LCF was added to 25 mL of solution. The control was a PDB solution containing AFB_1_ or OTA. The samples were incubated for 3 days in a thermostat at 25 °C. The samplings were done after inoculation of LCF (T0) and after 3 days of incubation (T3). The test was done in 3 replicates.

### 5.7. Mycotoxin Inhibition Assay with S. commune Unrefined Extract (LCF)

The unrefined extract of the exopolysaccharide *S. commune* (LCF) at a concentration of 1% *w/v* (data not shown) and 2% *w/v* was added to PDB (0.5 mL) and used as a nutrient medium for the cultivation of *A. flavus* and *A. carbonarius.* The nutrient medium with or without exopolysaccharides (control), was inoculated with a conidial suspension of *A. flavus* or *A. carbonarius* in a concentration of 10^6^ conidia in 0.2 mL of sterilized distilled water. The samples were incubated for up to 6 days at 25 °C in dark conditions as described by Parroni and collaborators [42].

### 5.8. Analysis of Monosaccharides in pLCF and sLCF Fractions

Trimethylsilyl methyl glycosides were obtained by derivatization with the reagent Sylon™ HTP (Merck Life Science S.r.l., Milano, Italy) after methanolysis of the polysaccharide with 3 M HCl in methanol at 85 °C for 16 h [43]. Analytical gas–liquid chromatography was performed on an Agilent Technologies 6850 gas chromatograph equipped with a flame ionization detector and using He as carrier gas. An HP-1 capillary column (Agilent Technologies Italia S.p.A., Milano, Italy, 30 m) was used to separate trimethylsilylated methyl glycosides (temperature program: 150–280 °C at 3 °C/min).

### 5.9. Purification of S. commune Unrefined Polysaccharide (LCF)

Unrefined lyophilized liquid culture filtrate (LCF) was precipitated with ethanol (VWR, Randor, PA, USA) (99%), on ice, to remove the lipids. After the centrifugation (45 min/13,000 rpm at 4 °C, Z326K HERMLE, Gosheim, Germany), the pellet was resuspended with 1 mL of 10 mM potassium phosphate buffer (VWR, Randor, PA, USA) (pH = 7.5). After resuspension, proteinase K (Thermo Fisher Scientific, Waltham, MA, USA) was added to the sample and left overnight in a shaking water bath at 37 °C. During the night the sample turned to gel form. The sample was centrifuged (40 min/4000 rpm at 4 °C), however, most of the sample remained in gel form, while only a small part of the fraction remained in the supernatant. Therefore, the precipitate (gel form) (pLCF) was separated from the supernatant (sLCF) and separately subjected to dialysis (10 kDa, Sigma-Aldrich, St. Louis, MO, USA) against the water. After 2 days of dialysis, the samples were lyophilized and utilized for subsequent analysis.

### 5.10. Assays of the Different Variants of S. commune Polysaccharide (pLCF and sLCF) on Mycotoxin Inhibition

#### 5.10.1. Assay of the Semi purified Lyophilized (pLCF) Polysaccharide Fraction from S. commune on Aflatoxin B_1_ and Ochratoxin A Biosynthesis

The pLCF (375 mg) was rehydrated in 5 mL of H_2_O. To disrupt the triple helix of the schizophyllan structure, the gel was heated for 4h at 140 °C, as described by Zhang and collaborators [29]. After the heating treatment, the gel took the form of a viscous solution.

Semi purified exopolysaccharide (pLCF) at a concentration of 1% *w/v* and 2% *w/v* was added to PDB (0.5 mL) and used as a nutrient medium for the cultivation of *A. Flavus* or *A. carbonarius*. The nutrient medium was inoculated with a suspension of fungal conidia (*A. flavus* or *A. carbonarius*) in a concentration of 10^6^ conidia in 0.2 mL of sterilized distilled water. PDB inoculated with *A. flavus* or *A. carbonarius* was used as a control. The samples were incubated for 6 days at 25 °C in dark conditions.

#### 5.10.2. Assay of the Fraction of the Polysaccharide from S. commune (sLCF) on Ochratoxin A and Aflatoxin B_1_ Biosynthesis

The lyophilized supernatant sLCF was extracted from the polysaccharide *S. commune* at a concentration of 1% *w/v* (data not shown), and 2% *w/v* was added to PDB (0.5 mL) and used as a nutrient medium for the cultivation of *A. flavus* or *A. carbonarius.* The nutrient medium, with a dissolved fraction of sLCF, was inoculated with a conidia suspension of the two test fungi using the same procedure as for pLCF.

### 5.11. Assay of the Comparison between Commercial Schizophyllan and Semi-Purified Extract (pLCF) of S. Commune on Inhibition of Mycotoxin Synthesis

The inhibition of mycotoxin production in *A. Flavus* and *A. Carbonarius* was compared between the pLCF extract of *S. Commune* and commercial β-glucan Schizophyllan (Invivogen, Toulouse, France), using the microtiter plate assay described by Loncar and collaborators [15]. The mycotoxin analyses were performed, as described by Fanelli and collaborators [44] and Amezqueta and collaborators [45].

### 5.12. Ultraviolet Visible Spectrophotometry

The possible presence of proteins in both samples was checked by UV analysis (Ultraviolet Visible Spectrophotometer, Thermo Fisher Scientific, Waltham, MA, USA). The pLCF and sLCF were dissolved separately in 2.4 mL of H_2_O (sLCF was completely soluble in water) and both samples were diluted 100 times before analysis.

### 5.13. ^1^H NMR Spectroscopy

Five mg of the pLCF sample were exchanged three times with 99.9% D_2_O by lyophilization (Lyophilizer FD-10, Labfreez, Changshan, Hunan, China) and then dissolved in 0.6 mL of DMSO-d6 and 99.96% D_2_O in a 6:1 ratio. Two mg of the sLCF sample were exchanged twice with 99.9% D_2_O by lyophilization and then dissolved in 0.6 mL of 99.96% D_2_O. Spectra were recorded on a 500 MHz VARIAN spectrometer (Varian Cary*50 UV-Vis, Agilent, Santa Clara, CA, USA) operating at 70 °C for pLCF and 50 °C for sLCF. Chemical shifts are given in parts per million, using the internal Me_2_SO signal (^1^H = 2.53 ppm) for the pLCF ^1^H spectrum and using the residual HOD signal (^1^H = 4.50 ppm at 50 °C) for the sLCF ^1^H spectrum. The ^1^H NMR spectra were processed using MestreNova 9.2 software (Mestrelab Research, Santiago de Compostela, Spain).

### 5.14. Analysis of Antioxidant Enzyme Activities in A. flavus and A. carbonarius Treated and Nontreated with Lyophilized Semipurified Filtrates (pLCF) of S. commune

The activities of superoxide dismutase (SOD), catalase (CAT), and glutathione peroxidase (GPX) were analyzed in (50 mg) mycelia of *A. flavus* and *A. carbonarius* treated and nontreated with 2% *w/v* of pLCF of *S. commune*, as previously described at Reverberi and collaborators [28]. The antioxidant activities were reported as unit mg protein^−1^.

### 5.15. Quantification of the Proteins

The quantification of the proteins in LCF and pLCF was performed according to the Bradford method [46].

### 5.16. Statistics

The arithmetic mean was used as the mean value, while the standard deviation was used as an indicator of the dispersion around the arithmetic mean. The test of the difference in relation to the reference value was carried out using the *t*-test for one independent measurement where 100 (control) was used as the reference value. Examination of the difference between fungi was carried out using the *t*-test for independent measurements, and the differences were considered significant when the *p*-value was <0.001. The analysis was done in the statistical software STATISTICA 12, Tibco, Palo Alto, CA, USA.

## Figures and Tables

**Figure 1 toxins-15-00177-f001:**
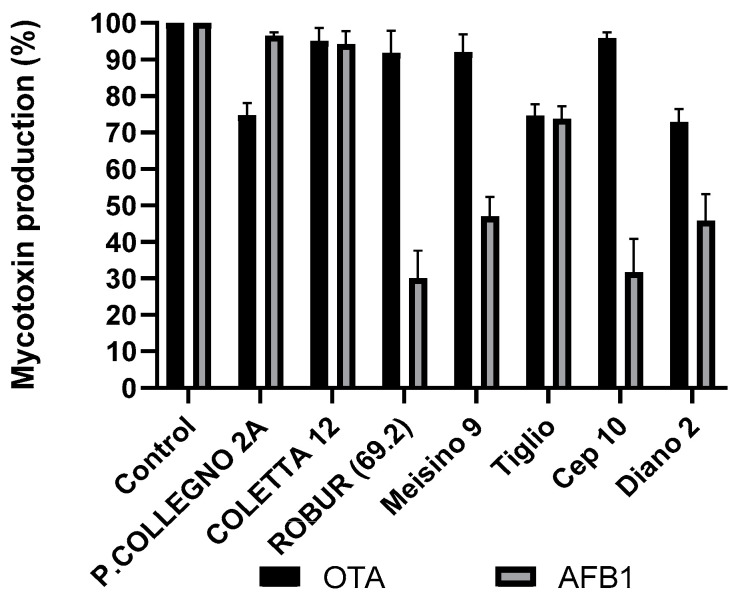
The effect of different isolates of mushroom *Ganoderma adspersum* lyophilized rough filtrates on ochratoxin A production in *A. carbonarius* and aflatoxin B_1_ production in *A. flavus* after 3 days incubation at 30 °C. Results are expressed as a percentage of mycotoxins produced relative to the control. The data are the mean ± SD of three determinations of five separate experiments.

**Figure 2 toxins-15-00177-f002:**
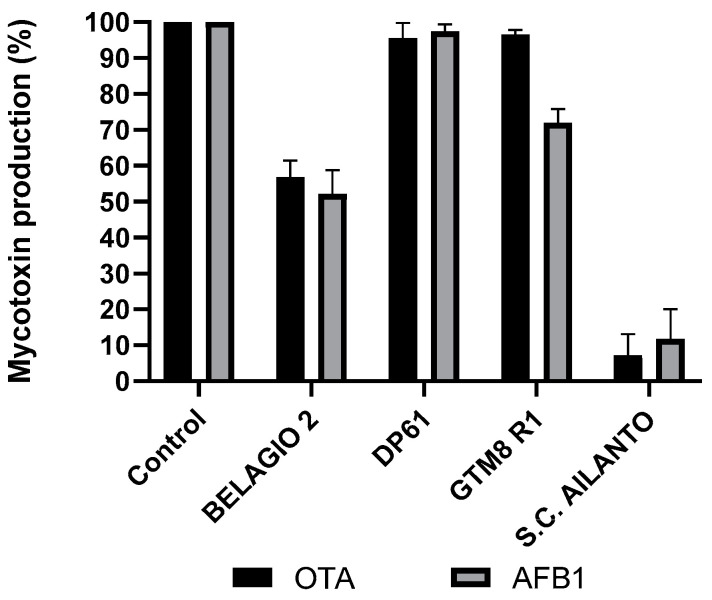
The effect of different isolates of the mushroom *S. commune* lyophilized filtrates on ochratoxin A production by *A. carbonarius* and aflatoxin B_1_ production by *A. flavus* after 3 days incubation at 30 °C. Results are expressed as a percentage of mycotoxin relative to the control. The data are the mean ± SD of three determinations of five separate experiments.

**Figure 3 toxins-15-00177-f003:**
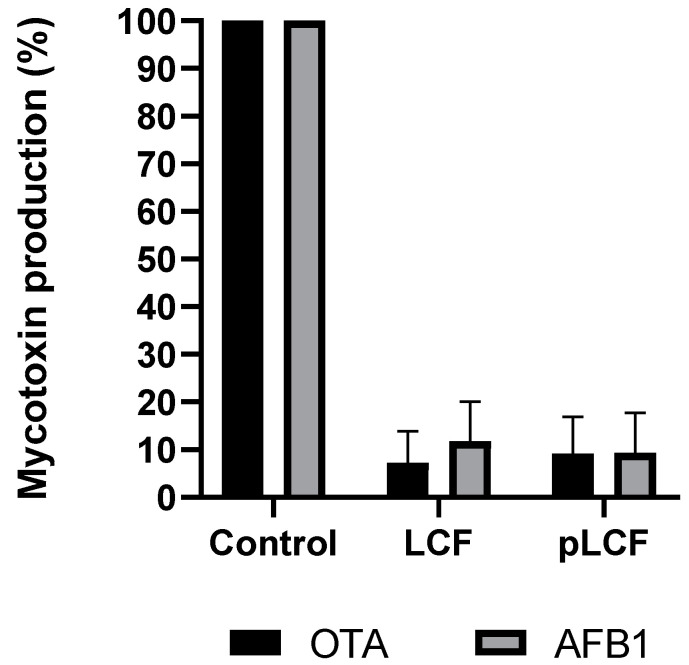
Comparison of the inhibition of biosynthesis of AFB_1_ and OTA in treatments with LCF and semipurified (pLCF) metabolite of S.C. Ailanto, compared with the control. Results are expressed as a percentage of mycotoxins produced relative to the control. The data are the mean ± SD of three different determinations of five separate experiments.

**Figure 4 toxins-15-00177-f004:**
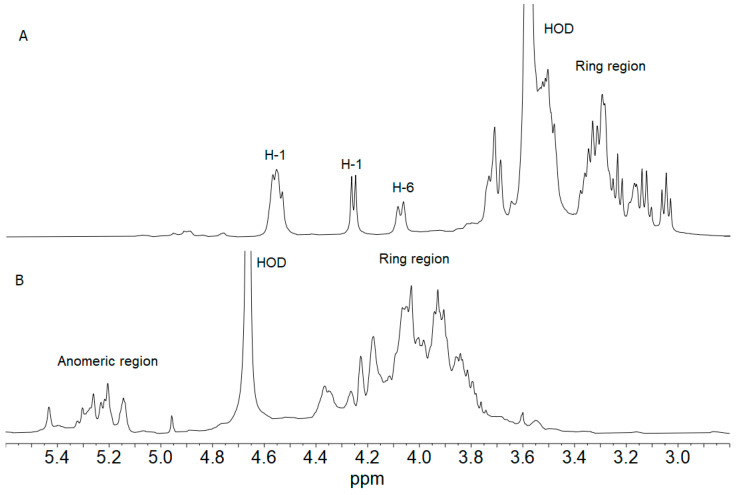
^1^H NMR spectra of fractions pLCF (**A**) recorded at 70 °C in deuterated DMSO:D_2_O in a 6:1 ratio and sLCF (**B**) recorded at 50 °C in D_2_O. Some assignments are indicated.

**Figure 5 toxins-15-00177-f005:**
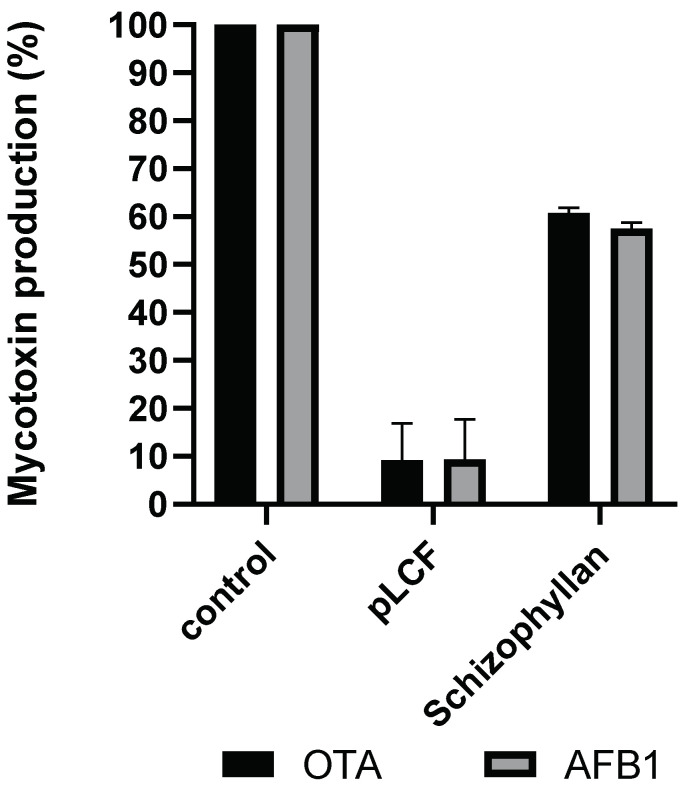
Comparison of the inhibition of biosynthesis of AFB_1_ and OTA in treatments with semi-purified pLCF (2% *w/v*) S. C. Ailanto metabolite and commercial polysaccharide Schizophyllan (2% *w*/*v*) compared to control. Results are expressed as a percentage of mycotoxins produced relative to the control. The data are the mean ± SD of three different determinations of five separate experiments.

**Figure 6 toxins-15-00177-f006:**
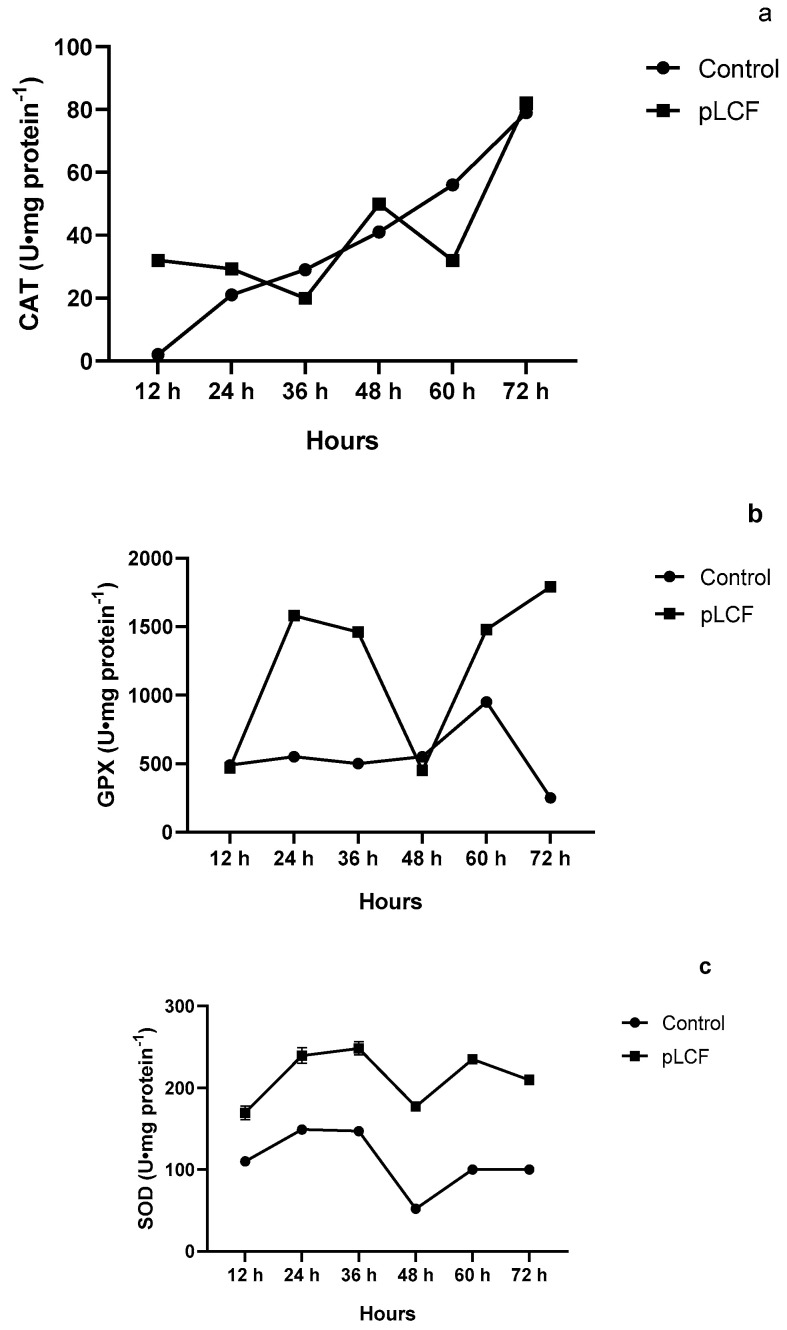
The activity of antioxidant enzymes CAT (**a**), GPX (**b**), and SOD (**c**), during the time course in the mycelium of *A. Flavus,* treated and not treated with semipurified filtrates pLCF of *S. commune*.

**Figure 7 toxins-15-00177-f007:**
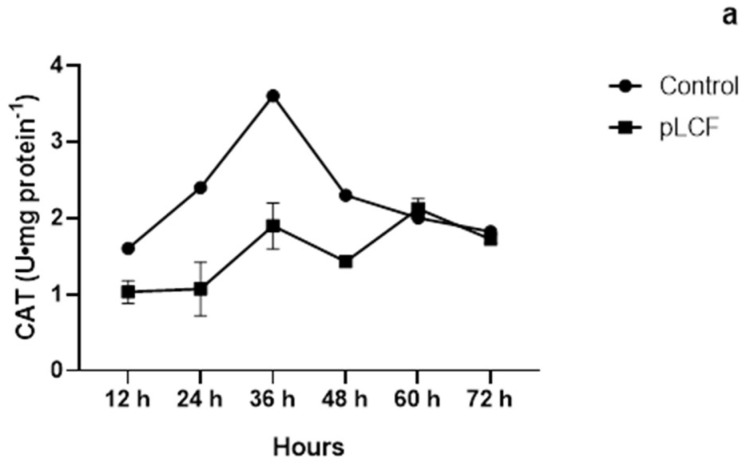
The activity of antioxidant enzymes CAT (**a**), GPX (**b**), and SOD (**c**), during the time course in the mycelium of *A. carbonarius,* treated and not treated with semipurified filtrates (pLCF) of *S. commune*.

**Table 1 toxins-15-00177-t001:** Inhibition rate (%) of LCF obtained from different fungal isolates on aflatoxin B_1_ and ochratoxin A production. Inhibitions < 50% were not considered significant for the purpose of this research and are reported as not significant (NS). Yield (DW) of LCFs is expressed in g • L^−1^. All data are the mean ± SD of determinations of ten separate experiments.

Code	Genera	Species	Inhibition of OTA(%) ± SD	T-Value (*p*)	Inhibition of AF (%) ± SD	T-Value (*p*)	Yield g • L^−1^
CF 16	*Agrocybe*	*aegerita*	6.64 ± 1.55	134.41 (<0.001)	70.18 ± 7.61	8.76 (0.001)	7.21 ± 0.50
CF 249	*Agrocybe*	*aegerita*	31.48 ± 3.75	40.82 (<0.001)	61.73 ± 7.38	11.60 (<0.001)	6.82 ± 0.52
CF 250	*Agrocybe*	*aegerita*	23.17 ± 3.33	51.66 (<0.001)	58.48 ± 8.24	11.27 (<0.001)	7.22 ± 0.48
CF 278	*Armillaria*	*mellea*	4.35 ± 2.60	82.24 (<0.001)	25.60 ± 2.58	64.56 (<0.001)	8.12 ± 0.46
CF 253	*Fomes*	*fomentarius*	37.06 ± 5.55	25.34 (<0.001)	19.91 ± 3.04	58.88 (<0.001)	9.32 ± 0.53
CF 255	*Fomes*	*fomentarius*	35.68 ± 4.61	31.17 (<0.001)	57.00 ± 7.92	12.14 (<0.001)	9.29 ± 0.50
CF 262	*Fomes*	*fomentarius*	7.24 ± 5.45	38.09 (<0.001)	45.70 ± 5.21	23.33 (<0.001)	9.45 ± 0.49
P.COLLEGNO 2A	*Ganoderma*	*adspersum*	25.30 ± 3.33	50.21 (<0.001)	3.45 ± 0.95	227.61 (<0.001)	9.11 ± 0.52
COLETTA 12	*Ganoderma*	*adspersum*	4.97 ± 3.69	57.62 (<0.001)	5.78 ± 3.57	59.06 (<0.001)	9.51 ± 0.60
ROBUR (69.2)	*Ganoderma*	*adspersum*	8.20 ± 6.20	33.11 (<0.001)	69.94 ± 7.59	8.86 (0.001)	9.59 ± 049
Meisino 9	*Ganoderma*	*adspersum*	8.00 ± 4.95	41.57 (<0.001)	53.01 ± 5.41	19.42 (<0.001)	7.62 ± 0.51
Tiglio	*Ganoderma*	*adspersum*	25.36 ± 3.12	53.56 (<0.001)	26.28 ± 3.48	47.33 (<0.001)	9.54 ± 0.54
Cep 10	*Ganoderma*	*adspersum*	4.14 ± 1.61	133.24 (<0.001)	68.32 ± 9.20	7.70 (0.002)	8.18 ± 0.57
Diano 2	*Ganoderma*	*adspersum*	27.10 ± 3.54	46.11 (<0.001)	54.20 ± 7.29	14.05 (<0.001)	9.89 ± 0.54
CF 17	*Ganoderma*	*lucidum*	4.60 ± 2.69	79.20 (<0.001)	5.11 ± 4.85	43.79 (<0.001)	9.56± 0.57
CF 223	*Ganoderma*	*lucidum*	7.31 ± 4.37	47.48 (<0.001)	20.02 ± 2.21	80.94 (<0.001)	9.12 ± 0.49
CF 264	*Ganoderma*	*lucidum*	28.82 ± 2.93	54.24 (<0.001)	54.70 ± 5.40	18.75 (<0.001)	9.53 ± 0.51
DP 1	*Ganoderma*	*resinaceum*	20.23 ± 2.64	67.55 (<0.001)	21.58 ± 2.94	59.58 (<0.001)	9.68 ± 0.52
PLATANO ROMA	*Ganoderma*	*resinaceum*	33.80 ± 3.76	39.33 (<0.001)	5.18 ± 4.79	44.23 (<0.001)	9.72 ± 0.48
P.VERDI 920	*Ganoderma*	*resinaceum*	41.27 ± 5.80	22.65 (<0.001)	25.28 ± 3.54	47.20 (<0.001)	7.88 ± 0.43
780.2	*Ganoderma*	*resinaceum*	42.06 ± 5.77	22.45 (<0.001)	25.10 ± 2.79	60.08 (<0.001)	9.82 ± 0.51
DP 63	*Ganoderma*	*resinaceum*	43.34 ± 5.14	24.67 (<0.001)	36.70 ± 5.28	26.81 (<0.001)	9.79 ± 0.46
ASPROMONTE 52	*Ganoderma*	*resinaceum*	53.36 ± 8.07	12.93 (<0.001)	31.29 ± 3.18	48.38 (<0.001)	8.89 ± 0.41
DP 23	*Ganoderma*	*resinaceum*	6.32 ± 4.92	42.62 (<0.001)	4.87 ± 5.80	36.65 (<0.001)	9.68 ± 0.50
C.TRENTO 43 B	*Ganoderma*	*resinaceum*	30.53 ± 4.21	36.86 (<0.001)	2.45 ± 1.10	198.08 (<0.001)	10.14 ± 0.47
P.VERDI 4A	*Ganoderma*	*resinaceum*	5.18 ± 3.74	56.64 (<0.001)	3.60 ± 1.56	138.07 (<0.001)	9.99 ± 0.52
G.R. 853	*Ganoderma*	*resinaceum*	6.43 ± 3.24	64.65 (<0.001)	28.20 ± 3.00	53.53 (<0.001)	9.53 ± 0.49
CF 258	*Ganoderma*	*resinaceum*	4.17 ± 1.09	196.13 (<0.001)	42.71 ± 6.38	20.09 (<0.001)	7.78 ± 0.51
CF 6	*Grifola*	*frondosa*	10.09 ± 3.28	61.24 (<0.001)	6.25 ± 4.14	50.67 (<0.001)	10.25 ± 0.41
CF 19	*Grifola*	*frondosa*	7.47 ± 3.48	59.44 (<0.001)	24.95 ± 3.13	53.68 (<0.001)	9.94 ± 0.51
CF 225	*Heterobasidion*	*annosum*	5.84 ± 3.09	68.06 (<0.001)	23.00 ± 3.00	57.33 (<0.001)	7.34 ± 0.49
CF 224	*Phellinus*	*pfefferi*	5.76 ± 4.28	49.27 (<0.001)	5.91 ± 4.53	46.40 (<0.001)	6.85 ± 0.59
CF 25	*Pleurotus*	*eryngii*	26.56 ± 3.23	50.85 (<0.001)	6.27 ± 4.16	50.40 (<0.001)	6.83 ± 0.50
CF 38	*Pleurotus*	*eryngii*	69.92 ± 9.66	6.96 (0.002)	23.38 ± 3.21	53.33 (<0.001)	7.48 ± 0.55
CF 28	*Polyporus*	*sulphureus*	24.92 ± 2.50	67.26 (<0.001)	2.53 ± 2.18	100.12 (<0.001)	9.58 ± 0.72
CF 280	*Trametes*	*hirsuta*	7.22 ± 5.65	36.69 (<0.001)	20.16 ± 2.08	86.00 (<0.001)	7.92 ± 0.54
T.V. 117	*Trametes*	*versicolor*	85.75 ± 11.76	2.71 (0.054)	89.00 ± 9.00	2.73 (0.052)	6.68 ± 0.50
BELAGIO 2	*Schizophyllum*	*commune*	43.20 ± 5.23	24.31 (<0.001)	47.91 ± 6.72	17.34 (<0.001)	9.81 ± 0.48
DP61	*Schizophyllum*	*commune*	4.56 ± 4.86	43.87 (<0.001)	2.59 ± 1.96	111.31 (<0.001)	7.92 ± 0.51
GTM8 R1	*Schizophyllum*	*commune*	3.46 ± 1.53	141.44 (<0.001)	28.12 ± 3.95	40.68 (<0.001)	8.76 ± 0.56
CF 18	*Stropharia*	*eugosoannulata*	2.76 ± 0.44	490.75 (<0.001)	35.34 ± 4.90	29.51 (<0.001)	9.92 ± 0.49
S.C. AILANTO	*Schizophyllum*	*commune*	92.76 ± 6.61	2.45 (0.070)	88.20 ± 8.21	3.22 (0.032)	8.59 ± 0.50

**Table 2 toxins-15-00177-t002:** Effect of LCF of *S. commune*. on AFB_1_ and OTA already present in the media after 3 days of incubation at 25 °C. Results are expressed as a percent of the initial concentration of mycotoxins, which was 10 g • L^−1^ The data are the mean ± SD of three different determinations of LCF that showed the presence of mannose, galactose, and glucose in molar ratio 8:2:1.

	AFB_1_		OTA	
Time (Days)	0	3	0	3
CONT	100	97.03 ± 2.87	100	96.14 ± 3.75
LCF	100	95.94 ± 3.85	100	96.48 ± 3.01

**Table 3 toxins-15-00177-t003:** Different mushroom isolates used for the experiment.

Code	Genus	Species
CF 16	*Agrocybe*	*Aegerita*
CF 249	*Agrocybe*	*Aegerita*
CF 250	*Agrocybe*	*Aegerita*
CF 278	*Armillaria*	*Mellea*
CF 253	*Fomes*	*fomentarius*
CF 255	*Fomes*	*fomentarius*
CF 262	*Fomes*	*fomentarius*
P.COLLEGNO 2A	*Ganoderma*	*adspersum*
COLLETTA 12	*Ganoderma*	*adspersum*
ROBUR (69.2)	*Ganoderma*	*adspersum*
Meisino 9	*Ganoderma*	*adspersum*
Tiglio	*Ganoderma*	*adspersum*
Cep.10	*Ganoderma*	*adspersum*
Diano 2	*Ganoderma*	*adspersum*
CF 17	*Ganoderma*	*lucidum*
CF 223	*Ganoderma*	*lucidum*
CF 264	*Ganoderma*	*lucidum*
DP1	*Ganoderma*	*resinaceum*
PLATANO ROMA	*Ganoderma*	*resinaceum*
P.VERDI 920	*Ganoderma*	*resinaceum*
780.2	*Ganoderma*	*resinaceum*
DP63	*Ganoderma*	*resinaceum*
ASPROMONTE 52	*Ganoderma*	*resinaceum*
DP23	*Ganoderma*	*resinaceum*
C.TRENTO 34 B	*Ganoderma*	*resinaceum*
P. VERDI 4A	*Ganoderma*	*resinaceum*
G.R. 853	*Ganoderma*	*resinaceum*
CF 258	*Ganoderma*	*resinaceum*
CF 6	*Grifola*	*frondosa*
CF 19	*Grifola*	*frondosa*
CF 225	*Heterobasidion*	*annosum*
CF 224	*Phellinus*	*pfefferi*
CF 25	*Pleurotus*	*eryngii*
CF 38	*Pleurotus*	*eryngii*
CF 28	*Polyporus*	*sulphureus*
BELLAGIO 2	*Schizophyllum*	*commune*
DP61	*Schizophyllum*	*commune*
GTM8 R1	*Schizophyllum*	*commune*
S.C. AILANTO	*Schizophyllum*	*commune*
CF 18	*Stropharia*	*rugosoannulata*
CF 280	*Trametes*	*hirsuta*
T.V. 117	*Trametes*	*versicolor*

## Data Availability

Not applicable.

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
