# Peer review of "The Effect of Mushroom Culture Filtrates on the Inhibition of Mycotoxins Produced by Aspergillus flavus and Aspergillus carbonarius"

_toxins, 2023, doi:10.3390/toxins15030177_

Round 1
Reviewer 1 Report
In the present study, authors screened 42 isolates of different ligninolytic mushrooms and their capacity to produce metabolites that simultaneously control the biosynthesis of both AFB1 and OTA. Further, they identified other active polysaccharides active in AFB1 and OTA inhibition. The author has addressed the main question of current research and is relevant in the
field. However, I have major comments on this manuscript.
- The author should include the Yield of each mushroom's unrefined lyophilized filtrates (LCFs)
- Why the author has used 2% exopolysaccharides for screening
- The author used semi-purified lyophilized (pLCF) polysaccharide fractions, and the author should provide compositions of the pLCF.
- Why has the author performed an antioxidant assay? What is the logic of this study?
- The author provided the proton NMR to confirm the active compound, but if they confirm the sugar composition of polysaccharides or more identification will improve and strengthen this manuscript.
- The author should add statistical analysis and improve all the figure presentation, especially labelling.
- The author should improve the discussion according to the results
Author Response
Response to Reviewer 1 Comments
Point 1: The author should include the Yield of each mushroom's unrefined lyophilized filtrates (LCFs)
Response 1: According to reviewer's suggestions the Yield of each mushroom is shown in Table 1
Point 2: Why the author has used 2% exopolysaccharides for screening
Response 2: The previous screening of the mushroom lyophilized filtrates in concentrations of 0,1%, 0,5%, 1% and 2 % (data not shown) have demonstrated the highest inhibiting effect on both mycotoxins at a concentration of 2%. Therefore, for this research, we have used the 2% concentration of all the mushroom lyophilized filtrates to better guarantee that the eventual active compound will act.
Point 3: The author used semi-purified lyophilized (pLCF) polysaccharide fractions, and the author should provide compositions of the pLCF.
Response 3: According to the reviewer's suggestions, the pLCF composition was performed. The results show that monosaccharides found are Glc with traces of mannose
Point 4: Why has the author performed an antioxidant assay? What is the logic of this study?
Response 4: Thanks for your interesting question. Our background demonstrated that there is a close connection between the onset of toxin synthesis and the redox status of the cell. Notably, we indicated that a pro-oxidant cell paves the way for opening toxin biosynthesis (REF to doi: 10.1007/s00253-010-2657-5). We demonstrated it for aflatoxins (doi: 10.1128/EC.00228-07), ochratoxin A (doi: 10.1007/s00253-012-3985-4) and patulin (doi: 10.1016/j.ijfoodmicro.2011.11.021). In this frame we discovered that if opportunely modulated we can switch off the synthesis of such mycotoxins by "simply" instaure a pro-reductive status in the cell. This could be done by using antioxidant or pro-antioxidant such as mushrooms' exopolysaccharides (e.g. Tramesan; doi: 10.1371/journal.pone.0171412). Henceforth, in this study on a novel exo-polysaccharide derived from Schizophyllum we aim at investigating for a putative pro-antioxidant ability of this as was for Tramesan.
Point 5: The author provided the proton NMR to confirm the active compound, but if they confirm the sugar composition of polysaccharides or more identification will improve and strengthen this manuscript.
Response 5: The GLC analysis of the active fraction pLCF and of sLCF was performed according to reviewer's suggestions. The two samples are different and the results were added in the manuscript. Moreover, 1H NMR spectrum of pLCF was recorded in 0.6 mL of DMSO-d6 and 99.96% D2O in 6:1 ratio and the spectrum showed that it is very similar or identical to schizophyllan. These data were added in the results paragraph.
Point 6: The author should add statistical analysis and improve all the figure presentation, especially labelling.
Response 6: The images have been improved according to the reviewer's suggestion. A description of the statistical analysis is also included in the text:
The use of graphic presentation methods presents the biosynthesis and inhibition of mycotoxins in the observed fungi (A. flavus and A. carbonarius), as well as the change over time (this change is in the last few graphs). The arithmetic mean is used as the mean value, while the standard deviation is used as an indicator of the dispersion around the arithmetic mean. The test of the difference in relation to the reference value is carried out using the T-test for one independent measurement where 100 (control) is used as the reference value.
Point 7: The author should improve the discussion according to the results
Response 7: The discussion is improved according to the results
Reviewer 2 Report
The manuscript investigated the effect of 42 isolates of mushroom culture filtrates on the inhibition of AFB1 and OTA roduced by Aspergillus flavus and Aspergillus carbonarius, and identify active polysaccharides active in AFB1 and OTA inhibition. The study designs well and obtain some interesting results. However, some comments for the manuscript need to be addressed before it can be accepted.
1. In Title: “Preliminary research on the” are redundant words, deleting these words has no negative influence on the presentation of the Tittle, and would make it clearer and more concise.
2. In the section of Introduction (lines 44-49), the prevention of fungal growth and/or mycotoxin control using chemical compounds (pesticides and fungicides) was introduced. Except for the detoxification means of AFB1, other ways to detoxify, such as animal function regulator (curcumin), enzyme, were not introduced. It is necessary to briefly introduce the detoxify means in the Introduction section, and allow me to suggest the following publication to be cited herein, please.
Wang, Y.; Liu, F.; Zhou, X.; Liu, M.; Zang, H.; Liu, X.; Shan, A.; Feng, X. Alleviation of Oral Exposure to Aflatoxin B1-Induced Renal Dysfunction, Oxidative Stress, and Cell Apoptosis in Mice Kidney by Curcumin. Antioxidants 2022, 11, 1082. https://doi.org/10.3390/antiox11061082.
Zhou, Z.; Li, R.; Ng, T.B.; Lai, Y.; Yang, J.; Ye, X. A New Laccase of Lac 2 from the White Rot Fungus Cerrena unicolor 6884 and Lac 2-Mediated Degradation of Aflatoxin B1. Toxins 2020, 12, 476. https://doi.org/10.3390/toxins12080476.
3. Line 70: “other active polysaccharides active in”, It's a little cumbersome to use the word “active” two times in the sentence.
4. Lines 73-80: It is more appropriate to move the paragraph to the “Discussion” section.
5. In Table 1, the meaning of numbers presented are confusing. The number values of inhibition of OTA and AF should all be percentages and less than 100 (100%), why are many values far greater than 100. In addition, should 0,001 be 0.001?
6. Lines 100-101: “One isolate of G adspersum, COLLETTA 12, showed no significant inhibition (p>0,001) (it should be p>0.001) of both mycotoxins”. But as shown in Table 1, p<0.001. which one is right?
7. Line 107: “isolate S. C. Alianto” should be “lyophilized filtrates of isolate S. C. Alianto”.
8. In Figure 2: “BELAGIO 2” and “S.C. AILANTO” shown as “BELAGI...”and S.C...” is incomplete, and rectify them please.
9. Line 141: “PDB”, give its full name “Potato Dextrose Broth” when the abbreviation was first appeared.
10. Line 166: “2.4.1.” should be “2.4”.
11. Line 167: “1H NMR” should be “1H NMR”.
12. Figure 5: panel A of the figure is missing, supplement it please. In addition, add space between “sLCF” and “recorded” in the captions of Figure 5.
13. In the captions of Figure 6, “scleroglucan”, but in line 169, “schizophyllan”, which is right?
14. In Figure 7, the legend of the column of group 3: “Schizophyllum” should be “Schizophyllan”.
15. On the ordinates of panel a, b, c of the Figures 8 and 9, representation of units of CAT, GPX and SOD need to be improved, for instance, changed to CAT (U•mg protein-1), GPX (U•mg protein-1) and SOD (U•mg protein-1)
16. Lines 282 and 283: “4100, 000” and “50,000” lack of unit, add unit please.
17. Line 322: the word “This” should be “It”.
18. Lines 364 and 375: it is better to replace (a), (b) with the serial numbers of 5.4.1, 5.4.2, respectively.
19. Line 417: the serial number “5.7.1.” should be “5.7.2”.
20. Lines 408, 434 and 445: “H20” should be ““H2O”. “D20” should be ““D2O”.
21. Line 432: add space between words, please.
22. In statistics, p<0.05 is considered statistically significant. But in this study, the differences were considered significant when the p-value was <0.001 (as shown in lines 431 and 464). Why not use the value p<0.05?
Author Response
Response to Reviewer 2 Comments
Point 1: In Title: “Preliminary research on the” are redundant words, deleting these words has no negative influence on the presentation of the Tittle, and would make it clearer and more concise.
Response 1: A very insightful comment. The title has been changed according to reviewer's suggestion.
Point 2: In the section of Introduction (lines 44-49), the prevention of fungal growth and/or mycotoxin control using chemical compounds (pesticides and fungicides) was introduced. Except for the detoxification means of AFB1, other ways to detoxify, such as animal function regulator (curcumin), enzyme, were not introduced. It is necessary to briefly introduce the detoxify means in the Introduction section, and allow me to suggest the following publication to be cited herein, please.
Wang, Y.; Liu, F.; Zhou, X.; Liu, M.; Zang, H.; Liu, X.; Shan, A.; Feng, X. Alleviation of Oral Exposure to Aflatoxin B1-Induced Renal Dysfunction, Oxidative Stress, and Cell Apoptosis in Mice Kidney by Curcumin. Antioxidants 2022, 11, 1082. https://doi.org/10.3390/antiox11061082.
Zhou, Z.; Li, R.; Ng, T.B.; Lai, Y.; Yang, J.; Ye, X. A New Laccase of Lac 2 from the White Rot Fungus Cerrena unicolor 6884 and Lac 2-Mediated Degradation of Aflatoxin B1. Toxins 2020, 12, 476. https://doi.org/10.3390/toxins12080476
Response 2: In the section of Introduction (lines 44-49) detoxification means of other ways to detoxify AFB1, such as animal function regulator (curcumin), and enzyme, are introduced according to reviewer instructions. The previously mentioned papers are also cited
Point 3: Line 70: “other active polysaccharides active in”, It's a little cumbersome to use the word “active” two times in the sentence.
Response 3: The word active has been removed as per your suggestion
Point 4: Lines 73-80: It is more appropriate to move the paragraph to the “Discussion” section.
Response 4: According to your suggestion, the paragraph has been moved to the “Discussion” section
Point 5: In Table 1, the meaning of numbers presented are confusing. The number values of inhibition of OTA and AF should all be percentages and less than 100 (100%), why are many values far greater than 100. In addition, should 0,001 be 0.001?
Response 5: The p value has been corrected according to your instructions from 0,001 to 0.001
We did not find a value that exceeds 100% . The t-ratio is a test value that is not directly related to the percentage, but is related to the inhibition ratio and the standard error of the inhibition. At low values the standard values of the ratios take exstremly large values as we have here. Moreover, this further confirms the presence of a difference because the standard errors of the data are low.
Point 6: Lines 100-101: “One isolate of G adspersum, COLLETTA 12, showed no significant inhibition (p>0,001) (it should be p>0.001) of both mycotoxins”. But as shown in Table 1, p<0.001. which one is right?
Response 6: The table presents the exact values. In the lines 100-101 we had error in writing.
The sentence was corrected according to the reviewer's suggestions.
Point 7: Line 107: “isolate S. C. Alianto” should be “lyophilized filtrates of isolate S. C. Alianto”.
Response 7: "isolate S. C. Alianto" was corrected according to reviewer's suggestions to "lyophilized filtrates of isolate S. C. Alianto".
Point 8: In Figure 2: “BELAGIO 2” and “S.C. AILANTO” shown as “BELAGI...”and S.C...” is incomplete, and rectify them please.
Response 8: The graph has been corrected according to reviewer's suggestions.
Point 9: Line 141: “PDB”, give its full name “Potato Dextrose Broth” when the abbreviation was first appeared.
Response 9: The PDB is listed as Potatoe Dextrose Broth per reviewer's suggestions.
Point 10: Line 166: “2.4.1.” should be “2.4”.
Response 10: The number has been corrected according to reviewer's suggestions.
Point 11: Line 167: “1H NMR” should be “1H NMR”.
Response 11: The “1H NMR” has been corrected to “1H NMR”.
Point 12: Figure 5: panel A of the figure is missing, supplement it please. In addition, add space between “sLCF” and “recorded” in the captions of Figure 5.
Response 12: The space between “sLCF” and “recorded” in the captions of Figure 5 has been added
Point 13: In the captions of Figure 6, “scleroglucan”, but in line 169, “schizophyllan”, which is right?
Response 13: As per reviewer's suggestion the word “scleroglucan” was replaced with the word schizophyllan“ - which is the correct information“
Point 14: In Figure 7, the legend of the column of group 3: “Schizophyllum” should be “Schizophyllan”.
Response 14: The coulumn of group 3 “Schizophyllum” is corrected to “Schizophyllan”
Point 15: On the ordinates of panel a, b, c of the Figures 8 and 9, representation of units of CAT, GPX and SOD need to be improved, for instance, changed to CAT (U•mg protein-1), GPX (U•mg protein-1) and SOD (U•mg protein-1)
Response 15: On the ordinates of panel a, b, c of the Figures 8 and 9, representation of units have been improved to CAT (U•mg protein-1), GPX (U•mg protein-1) and SOD (U•mg protein-1)
Point 16: Lines 282 and 283: “4100, 000” and “50,000” lack of unit, add unit please.
Response 16: The sentence was corrected according to the reviewer's suggestion.
Point 17: Line 322: the word “This” should be “It”.
Response 17: According to reviewer's suggestion, the word “This” was replaced with the word “It”
Point 18: Lines 364 and 375: it is better to replace (a), (b) with the serial numbers of 5.4.1, 5.4.2, respectively.
Response 18: According to the reviewer's instructions, (a) and (b) were replaced with the serial numbers of 5.4.1, 5.4.2, respectively.
.Point 19: Line 417: the serial number “5.7.1.” should be “5.7.2”.
Response 19: According to the reviewer's instructions the serial number “5.7.1.” is replaced with “5.7.2.”.
Point 20: Lines 408, 434 and 445: “H20” should be ““H2O”. “D20” should be ““D2O”.
Response 20: Lines 408, 434 and 445 have been corrected according to the reviewer's instructions.
Point 21: Line 432: add space between words, please.
Response 21: According to the reviewer's instructions the space between words is added.
Point 22: In statistics, p<0.05 is considered statistically significant. But in this study, the differences were considered significant when the p-value was <0.001 (as shown in lines 431 and 464). Why not use the value p<0.05?
Response 22: Within the theoretical explanation of the results, it is required to be consistent. Therefore, every number less than 0.001 is less than 0.050. In all the places where it is written <0.001, it can be written that it is less than 0.050.
Lowe p-value gives the highest statistical value. We have used higher statistical power and wanted to be 99,9% confident which means that our results are not due to chance.
Reviewer 3 Report
The work of Preliminary research on the effect of mushroom culture filtrates on the inhibition of mycotoxins produced by Aspergillus flavus and Aspergillus carbonarius is very interesting. The manuscript is based on a fairly good experimental design. However, the manuscrpt seems to be written by multiple authors but not well-organised. It requires careful checking and making changes.
L39-49. Please re-write the sentence.
L73-80. The authors described the aim of this study. It would be better to move the fragments from the Result Section to the Introduction Section.
L96. All data are the mean ± SD of determinations of ten separate experiments ±SE. What dose you mean?
L108-128. These framents should not be present in the Result Section. Please just clearly present results in the Result Section but not discuss the findings in the Result Section. This also applies to the rest of the manuscript.
1H-NMR, 13C-NMR, 1H-1H COSY and HMBC should all be obtained to identify the structure of sLCF.
Where is the result of UV-Visable spectrum of pLCF and sLCF as described in MM5.9?
L215. SOD activity (Fig. 8 C) was significantly stimulated (p<0.001 at 36h)….However, no symbols were found to show statistical significance in the Figure. This also applies to the rest of the manuscript.
I suggest the authors to draw Figures with Graphpad Prism but not with EXCEL.
L93 and L328. There are two Table 1. Please make changes
Please carefully check the reference lists. R7 and R8 are same. R19 and R20 are same.
Author Response
Response to Reviewer 3 Comments
Point 1: L39-49. Please re-write the sentence.
Response 1: The sentence was modified according to the reviewer's suggestions.
Point 2: L73-80. The authors described the aim of this study. It would be better to move the fragments from the Result Section to the Introduction Section.
Response 2: The aim of the study has been transferred from the Result section to the Introduction section according to the reviewer's suggestions.
Point 3: L96. All data are the mean ± SD of determinations of ten separate experiments ±SE. What dose you mean?
Response 3: Results are the mean of ± SD. SE was error in writing.
Point 4: L108-128. These framents should not be present in the Result Section. Please just clearly present results in the Result Section but not discuss the findings in the Result Section. This also applies to the rest of the manuscript.
Response 4: The results section was revised according to the reviewer's suggestions.
Point 5: 1H-NMR, 13C-NMR, 1H-1H COSY and HMBC should all be obtained to identify the structure of sLCF.
Where is the result of UV-Visable spectrum of pLCF and sLCF as described in MM5.9?
Response 5: the sLCF fraction was found to be NOT active in the biological tests performed. Therefore, its structure is not of primary interest for this manuscript and we think that the data obtained on the effect of mushroom culture filtrates on the inhibition of mycotoxins can stand alone for the publication. Moreover, the 1H NMR spectrum of pLCF was recorded in 0.6 mL of DMSO-d6 and 99.96% D2O in 6:1 ratio and it resulted to be very similar or identical to schizophyllan. These data were added in the results paragraph. Further 2D NMR spectroscopy experiments will be considered in order to undoubtedly establish its primary structure.
A figure with the UV spectra is present in the Supplementary file
Point 6: L215. SOD activity (Fig. 8 C) was significantly stimulated (p<0.001 at 36h)….However, no symbols were found to show statistical significance in the Figure. This also applies to the rest of the manuscript.
Response 6: In the theoretical explanation of the results, every number less than 0.001 is less than 0.050. Lowe p-value gives the highest statistical value. We have used higher statistical power and wanted to be 99,9% confident which means that our results are not due to chance.
Point 7: I suggest the authors to draw Figures with Graphpad Prism but not with EXCEL.
Response 7: According to the suggestions, the graphs were made in the GraphArt program
Point 8: L93 and L328. There are two Table 1. Please make changes
Response 8: Since we have now replaced one graph with a table, there are 3 tables in the text and they are all marked exactly according to the reviewer's instructions
Point 9: Please carefully check the reference lists. R7 and R8 are same. R19 and R20 are same.
Response 9:
- Reference R8 has been substituted with the reference: Tao, Y., Xie, S., Xu, F., Liu, A., Wang, Y., Chen, D., Pan, Y., Huang, L., Peng, D., Wang, X., Yuan, Z. Ochratoxin A: Toxicity, oxidative stress and metabolism. Food Chem. Toxicol. 2018, 112, 320-331. https://doi.org/10.1016/j.fct.2018.01.002
- The R20 reference has been deleted
Round 2
Reviewer 1 Report
The authors have satisfactorily responded to all comments and made the necessary changes to the manuscript.
Reviewer 2 Report
All my comments have been addressed in the revised manuscript, and I consider it can be accepted.
Reviewer 3 Report
The manuscript has been improved and could be considered for publication in Toxins